# Transcriptome Analysis of a Cotton Cultivar Provides Insights into the Differentially Expressed Genes Underlying Heightened Resistance to the Devastating Verticillium Wilt

**DOI:** 10.3390/cells10112961

**Published:** 2021-10-30

**Authors:** He Zhu, Jian Song, Nikhilesh Dhar, Ying Shan, Xi-Yue Ma, Xiao-Lei Wang, Jie-Yin Chen, Xiao-Feng Dai, Ran Li, Zi-Sheng Wang

**Affiliations:** 1National Cotton Industry Technology System Liaohe Comprehensive Experimental Station, The Cotton Research Center of Liaoning Academy of Agricultural Sciences, Liaoning Provincial Institute of Economic Crops, Liaoyang 111000, China; zhuhe19840204@163.com (H.Z.); shanying_sy@163.com (Y.S.); wangxiaolei0509@126.com (X.-L.W.); 2State Key Laboratory for Biology of Plant Diseases and Insect Pests, Institute of Plant Protection, Chinese Academy of Agricultural Sciences, Beijing 100193, China; songjian_01@126.com (J.S.); 13333227788@163.com (X.-Y.M.); chenjieyin@caas.cn (J.-Y.C.); daixiaofeng_caas@126.com (X.-F.D.); 3Department of Plant Pathology, University of California, Davis, c/o United States Agricultural Research Station, Salinas, CA 93905, USA; ndhar@ucdavis.edu

**Keywords:** cotton, disease resistance, *Verticillium dahliae* (*V. dahliae*), RNA-seq, transcriptomics, gene regulatory networks, verticillium wilt (VW)

## Abstract

Cotton is an important economic crop worldwide. Verticillium wilt (VW) caused by *Verticillium dahliae* (*V. dahliae*) is a serious disease in cotton, resulting in massive yield losses and decline of fiber quality. Breeding resistant cotton cultivars is an efficient but elaborate method to improve the resistance of cotton against *V. dahliae* infection. However, the functional mechanism of several excellent VW resistant cotton cultivars is poorly understood at present. In our current study, we carried out RNA-seq to discover the differentially expressed genes (DEGs) in the roots of susceptible cotton *Gossypium hirsutum* cultivar Junmian 1 (J1) and resistant cotton *G.*
*hirsutum* cultivar Liaomian 38 (L38) upon *Vd*991 inoculation at two time points compared with the mock inoculated control plants. The potential function of DEGs uniquely expressed in J1 and L38 was also analyzed by GO enrichment and KEGG pathway associations. Most DEGs were assigned to resistance-related functions. In addition, resistance gene analogues (RGAs) were identified and analyzed for their role in the heightened resistance of the L38 cultivar against the devastating *Vd*991. In summary, we analyzed the regulatory network of genes in the resistant cotton cultivar L38 during *V. dahliae* infection, providing a novel and comprehensive insight into VW resistance in cotton.

## 1. Introduction

Plants are under constant attack from various pests including bacteria, fungi, oomycetes, nematodes, and insects. As plant are unable to escape attacking pathogens by moving to a more favorable environment, they possess efficient defense mechanisms to protect themselves from disease, such as physical barriers and production of antimicrobial compounds [1,2]. Thus, plants have innate immunity against numerous potential pathogens in the environment [1,2,3]. Innate immune responses in plants can be broadly classified into two categories, i.e., PAMP-triggered immunity (PTI) and effector-triggered immunity (ETI) [3]. PTI is the first line of a plant’s active defense response upon microbial perception. It is initiated upon recognition of conserved microbial molecular signatures by plant cell-surface receptors, and induction of PTI is associated with activation of MAP kinase signaling leading to transcriptional induction of pathogen-responsive genes, production of reactive oxygen species (ROS), and deposition of callose to reinforce the cell wall at sites of infection, all of which contribute to restrict microbial growth [4]. Moreover, a series of plasma membrane-bound receptors were employed for regulation of immunity, including receptor-like kinases (RLKs) and receptor-like proteins (RLPs) [5,6,7]. The second line of defense response is regulated by intracellular R proteins of the host which are designed to recognize specific effectors secreted by the pathogen, and is hence designated as the ETI [1,3]. Notable motifs common to the *R* genes include the nucleotide-binding site (NBS), leucine-rich repeat (LRR), serine/threonine protein kinase domain (PK), drosophila toll domain, coiled-coil structure (CC), and transmembrane domain (TM) [8]. Moreover, in the last two decades, an increasing repertoire of plant signaling molecules actively regulating plant defense against a myriad of pathogen attack have been investigated, including plant hormones such as salicylic acid (SA), jasmonic acid (JA) and ethylene (ET) [9]. SA-dependent signaling is critical in establishing local and systemic bacterial resistance, ET-dependent signaling is crucial for the response to necrotrophic pathogens and mechanical wounding, while JA-dependent signaling is induced in response to mechanical wounding and herbivore predation [10].

Cotton (*Gossypium* spp.) is an important economic crop species cultivated worldwide. Verticillium wilt (VW) of cotton is a serious vascular disease that is detrimental to cotton yield and fiber quality, and is caused by the soil-born fungus *Verticillium dahliae* (*V. dahliae*). It is known that *V. dahliae* attacks host plants through their roots, generally takes around 2–4 days to reach the xylem vessels, and proliferates in the vascular to induce VW [11]. VW induced typical symptoms include vascular discoloration, leaf chlorosis, curling or necrosis, followed by defoliation and eventual wilt and plant death as the disease progresses [11]. In addition, as a soil borne pathogen, *V. dahliae* is difficult to control owing to its ability to survive long-term as microsclerotia in the soil and adapt to a broad host range to survive and propagate [12]. Breeding of resistant varieties is the effective and economic approach to prevent and control VW [13,14,15]. To this date, several VW resistance quantitative trait loci (QTLs) have been characterized based on development of high-throughput sequencing and have been shown to contribute to defense responses against VW [14,16,17]. In addition, several genes have been identified that contribute to defense responses against VW, including transcription factors, genes involved in plant hormone signaling network and *R* genes [18]. For example, the NBS-LRR genes *GbaNA1* [19] and *GhDSC1* [20] mediate cotton resistance against *V. dahliae*. Adenosine triphosphate (ATP)-binding cassette (ABC) genes have been identified that play a vital role in cotton against *V. dahliae* infection [21]. 12-oxo-phytodienoic acid reductases (OPRs) played important role in plant development and growth. The OPR gene family in cotton has been identified and *GhOPR9* has been found to be a positive regulator under *V. dahliae* infection which can modulate the expression of genes related to the JA pathway [22]. Calcium-dependent protein kinase GhCPK33 interact with GhOPR3 by phosphorylating it thereby negatively regulating cotton resistance to *V. dahlia* [23]. The WRKY transcription factor family plays a crucial role in plant defense against various pathogen infections. *GhWRKY70D13* negatively regulated cotton defense against *V. dahliae* through regulation of the genes involved in ET and JA biosynthesis and signaling pathways [24]. GhMAPK family members were also shown to play important roles in subtle regulation of cotton resistance to *V. dahlia* [25]. GhWAK7A, a cotton wall-associated kinase, mediated immunity response to *V. dahliae* by interacting with the chitin sensory receptors [26]. In addition, phosphate deficiency can enhance resistance of cotton against *V. dahliae* by activating JA biosynthesis and phenylpropanoid pathway [27]. 

The *Gossypium**hisutrum* cultivar Liaomian series bred by the Research of Economic Crops in Liaoning Province has received high praise from the domestic and international community for selection and breeding of Verticillium wilt resistance into cotton varieties. Guannong No. 1 was systematically bred from Mokpo 113–2 which is a self-bred and early-maturing variety of upland cotton from 1938. From the late 1950s to the early 1960s, the earliest VW-resistant varieties of Liaomian No. 1 and Liaomian No. 2 were bred from Guannong No. 1 cultivar. Whereafter, a series of cotton varieties have been cultivated, including Liaomian No. 10, integrating high yield, disease resistance, early maturity, high quality, and high resistance to Fusarium wilt and Verticillium wilt. Thus, after nearly a hundred years of development, the Liaoning Provincial Economic Crop Research Institute has bred 46 cotton varieties to 2019, that have been promoted and widely planted in cotton areas in China. Therefore, whether Liaomian cotton is used as an early maturity resistance source material or a disease-resistant variety, it has had a far-reaching influence in the selection of extra-early maturity and disease-resistant varieties, and has strongly promoted the progress and development of disease-resistant cotton breeding in China. However, the resistant function underlying the excellent resistance of Liao cotton to VW has not been clearly defined and hence is poorly understood. 

RNA-seq has been widely applied to monitor the gene expression profiles and describe the response characteristics of cotton roots under *V. dahliae* infection [17,28,29,30,31,32], and has a great potential to enrich our understanding of how cotton has elaborated various defense mechanisms against pathogen attacks. Therefore, in the current work, we conducted RNA-seq to discover the genes expressed in the roots of *Gossypium*
*hirsutum* resistant cultivar Liaomian No. 38 (L38) and *G. hirsutum* susceptibility cultivar Junmian No. 1 (J1) response to *V. dahliae* infection, respectively. The aim of this study was to discover the differentially expressed genes after *V. dahliae* infection to identify the potential resistant network of L38. The significantly different expressed genes (DEGs) were classified and divided into six groups, including J1-early, J1-late, J1-sustained, L38-early, L38-late, and L38-sustained. The potential function of DEGs in these groups were analyzed and putative association with sectors associated with disease resistant pathways, such as phenylpropanoid biosynthesis, plant hormone signal transduction, Mitogen-activated protein kinase (MAPK) signaling pathway and plant-pathogen interaction were revealed from our study. In addition, the resistance gene analogues (RGAs) that displayed differential expression were further analyzed. Our findings suggest that the expression of several resistance-related genes was induced in the resistant cotton cultivar L38 in a manner different from the susceptible cultivar J1 during *V. dahliae* infection, thus providing an insight into the genomic networks at play that result in the VW resistance of the L38 cotton cultivar.

## 2. Materials and Methods

### 2.1. Plant Growth Conditions and Fungal Pathogen Inoculations

The susceptible cotton *G. hirsutum* cultivar Junmian 1 (J1) and resistant cotton *G.*
*hirsutum* cultivar Liaomian 38 (L38) were grown and maintained in a greenhouse at 28 °C under 16 h light/8 h dark photoperiod. Moreover, L38 and J1 were planted at a disease nursery in Liaoyang, located in northeast of China (41°16′ north latitude, 123°12′ east longitude).

The highly virulent *V. dahliae* strain Vd991 [33] was cultured in complete medium (CM) at 25 °C for about 5 days by 200 rpm. Conidia were harvested by centrifugation and washed with sterile water. The final concentration was adjusted to 5 × 10^6^ conidia/mL which was then used for inoculating cotton seedlings [19]. Roots of the seedlings were immersed in the conidial suspension for 30 min and then planted into soil. The roots were sampled at 24 h and 72 h post inoculation. The cotton roots without inoculation were sampled at 0 h. All samples were immediately frozen in liquid nitrogen and stored at −80 °C until further use.

The disease index (DI) of Verticillium wilt was obtained according to the following formula:$$\mathrm{DI}=\frac{\sum \left(n\;\times \;\mathrm{number}\;\mathrm{of}\;\mathrm{plants}\;\mathrm{at}\;\mathrm{level}\;n\right)}{4\;\times \;\mathrm{the}\;\mathrm{number}\;\mathrm{of}\;\mathrm{total}\;\mathrm{plants}\times 100}$$

The index, comprising five grades (0, 1, 2, 3, and 4), was used for assigning infected plants. Grade 0 indicates that the plant is healthy and has no disease symptoms, grades 1 to 4 represent the typical yellowing and wilting observed in 0–25%, 25–50%, 50–75%, and 75–100% leaves of investigated plants, respectively.

### 2.2. RNA Extraction, Library Construction, and RNA Sequencing Analysis

Both J1 and L38 root samples were inoculated with Vd991 after 24 h and 72 h, in addition, the samples without inoculation and collected just prior to the treatments were designated as 0 h. For each experiment three technical replicates were performed with each replicate including 6 seedlings. Therefore, a total of 18 root samples were chosen for RNA extraction. Total RNA samples of cotton roots were extracted using Plant RNA Purification Kit (Tiangen, Beijing, China). The RNA concentration and purity were measured by NanoDrop 2000 spectrophotometer (ThermoFisher, Waltham, MA, USA). The RNA integrity was checked by agarose gel electrophoresis. Genomic DNA was removed by DNase treatment.

RNA-Seq library preparation and sequencing were performed using the standard Illumina protocols (Collibri™ Stranded RNA Library Prep Kit for Illumina™ Systems, Thermo, Waltham, MA, USA). Briefly, mRNAs were enriched from 1.5 μg total RNA by using magnetic beads with Oligo (dT) and used as templates to synthesize the first stranded cDNAs with random hexamers and produce double-stranded by using DNA Polymerase I. The products were enriched with PCR to create the final cDNA libraries after addition of A tail and ligation with sequencing adapters. Strand-specific sequencing was performed on an Illumina HiSeq X-Ten by BGI (BGI-genomics, Shenzhen, China), which generated 150 bp paired-end reads. The raw reads were filtered by a series of steps, including removal of lower quality read, adapters, higher content base of N reads, to obtain clean reads. The clean reads were mapped onto reference genome of *Gossypium hirsutum* (NCBI_GCF_000987745.1_ASM98774v1) by HISAT (Hierarchical Indexing for Spliced Alignment of Transcripts) [34]. The data presented in this article have been deposited in the National Center for Biotechnology Information (NCBI) Sequence Read Archive (http://www.ncbi.nlm.nih.gov/sra/) (26 October 2021). The accession number is currently pending submission until acceptance of the manuscript.

### 2.3. Identification of Differentially Expressed Genes (DEGs)

A total of six groups of cotton root sample (J1-0, J1-24, J1-72, L38-0, L38-24, and L38-72, response to the inoculation time points) were used for DEGs analysis. Significantly, DEGs were identified from four comparisons, including J1-24 compared with J1-0 (J1-24/0), J1-72 compared with J1-0 (J1-72/0), L38-24 compared with L38-0 (L38-24/0) and L38-72 compared with L38-0 (L38-72/0). Fragments per kilobase per million mapped fragments (FPKM) were calculated by RSEM and used to estimate the effects of sequencing depth and gene length on the mapped read counts. Foldchange in gene expression value was calculated by FPKM J1-24/FPKM J1-0, FPKM J1-72/FPKM J1-0, FPKM L38-24/FPKM L38-0, and FPKM L38-72/FPKM L38-0. DEseq2 was used to analyze DEGs in cotton under the criteria of a corrected Foldchange ≥2 and adjusted *p*-value ≤ 0.05 [35]. The heatmap of DEGs was clustered using the python package (seaborn.clustermap(parameters)). The cluster distance metric is ‘Euclidean’ and the cluster method is ‘ward’.

### 2.4. Gene Ontology Enrichment and KEGG Pathway Analysis

Gene ontology (GO) term enrichment analysis of DEGs was performed using WEGO (http://wego.genomics.org.cn/) (26 October 2021) in three terms, including cellular component, molecular functional and biological function, if the *p*-value < 0.05. Kyoto Encyclopedia of Genes and Genomes (KEGG) analysis was then performed in KEGG Mapper (https://www.kegg.jp/kegg/mapper.html) (26 October 2021).

### 2.5. Reverse Transcription and Quantitative PCR (RT-qPCR)

RNA aliquots of 2.0 µg were used for cDNA synthesis by the TranScript One-Step gDNA Removal and cDNA Synthesis SuperMix kit (Trans, Beijing, China). Quantitative PCR (qPCR) was performed using SYBR Green PCR Master Mix (TransGen Biotech, Beijing, China) on a CFX96 Touch Real-time PCR Detection System (Bio-Rad, http://www.bio-rad.com/) (26 October 2021). The relative quantification of RT-qPCR was measured by 2^−ΔΔCt^ analysis method [36]. The mRNA expression levels were normalized using cotton *18S* gene (*18S rRNA*). Three biological replicates were performed for each experiment, with three technical replicates. The specific primers used are listed in Appendix A.

### 2.6. Statistical Analysis

Student’s *t*-test was used to determine whether the RT-qPCR results were statistically different from two samples (* *p* < 0.05; ** *p* < 0.01). Duncan’s multiple range test was used for three samples (*p* < 0.05).

## 3. Results

### 3.1. Resistant Phenotype of L38 Cotton

L38 is a VW immune germplasm cotton, developed by the Liaoning Provincial Economic Crop Research Institute, China. To probe the resistance of the L38 cultivar to *V. dahliae*, seedlings of both L38 and VW susceptible J1 cultivars at four weeks were inoculated with *Vd*991. L38 plants showed highly resistant phenotype to *Vd*991 compared to J1 at 14 days post-inoculation (Figure 1A). As expected, the control treated plants of both J1 and L38 cultivars grew normally when they treated with water (Figure 1A). Moreover, the disease index of the resistant L38 cultivar was significantly lower than the susceptible J1 cultivar, consistent with the visual disease phenotypes (Figure 1B). More than 70% of J1 plants were detected at grade 4 which was equivalent to the highest disease severity in contrast to about only 10% of L38 plants with grade 4 disease index (Figure 1B). In addition, the disease symptoms were also confirmed at the disease nursery. L38 and J1 were planted at the Liaoyang disease nursery. L38 showed a highly resistant phenotype compared with J1 (Figure 1C). The disease index of L38 was also significantly lower than J1 detected in the disease nursery, consistent with disease phenotypes (Figure 1D). Above all, these results demonstrate that the L38 cultivar was significantly resistant to *V. dahliae* and showed enhanced VW resistance in comparison to the J1 cultivar.

### 3.2. RNA-Seq Was Conducted to J1 and L38 Cotton

Since, the cotton cultivar L38 showed an obvious *V. dahliae* resistant phenotype, we hypothesized that the observed phenotype was a direct result of altered expression of genes associated with *V. dahliae* resistance. To further explore the genetic network underlying the resistance of L38 cotton to *V. dahliae* infection, RNA-seq was performed. The roots of both L38 and J1 cotton plant cultivars were inoculated with the *V. dahliae* isolate Vd991 as described earlier [14]. Root samples were collected at 24 h and 72 h post-inoculation along with the uninoculated root samples at the start of the treatment (0 h). Therefore, a total of six samples were used for sequencing, including J1-0, J1-24, J1-72, L38-0, L38-24, and L38-72. Three independent biological replicates of the experiment were performed. All clean reads were mapped onto the *G. hirsutum* reference genome (NCBI_GCF_000987745.1_ASM98774v1), and more than 60% of clean reads were uniquely mapped to the genome of *G. hirsutum* in each sample file to ensure the credibility of the RNA-seq data (Appendix A). The expression of all genes in J1-0, J1-24, J1-72, L38-0, L38-24, and L38-72 is shown in Figure 2A. Based on hierarchical clustering using the FPKM values of all genes, it was found that the six samples could be classified into three groups (Figure 2A). DEGs of group 1 were highly expressed in J1 than in L38, while DEGs of group 3 were highly expressed in L38 than in J1 (Figure 2A). Most DEGs involved in group 2 comprised genes whose expression was high in both the J1 and L38 cultivars tested (Figure 2A).

### 3.3. Identification of Differentially Expressed Genes in L38 during Pathogen Inoculation

Differentially expressed genes (DEGs) were identified as *p* < 0.05 and fold change > 2.0 in J1-24h compared with J1-0h (J1-24/0), J1-72h compared with J1-0h (J1-72/0), L38-24 compared with L38-0h (L38-24/0), and L38-72h compared with L38-0h (L38-72/0) respectively. Thousands of DEGs were detected in the J1-24/0, J1-72/0, L38-24/0, and L38-72/0 groups. Venn diagrams of DEGs illustrate both commonly expressed and specifically expressed genes between the J1 and L38 cultivars. For up-regulated DEGs, 490 were specifically expressed in J1-24/0, 1097 were specifically expressed in J1-72/0, while 85 genes had sustained elevated levels both in J1-24/0 and J1-72/0 timepoints. Similarly, 578 genes were specifically expressed in L38-24/0, 2556 were specifically expressed in L38-72/0 group, while 771 genes had sustained expression levels in both L38-24/0 and L38-72/0 treatment groups following the *V. dahliae* inoculation (Figure 2B). For down-regulated DEGs, 385 were specifically expressed in J1-24/0, 698 were specifically expressed in J1-72/0, and 399 were consistently downregulated in both J1-24/0 and J1-72/0 treatment groups, 531 were specifically expressed in L38-24/0, 1991 were specifically expressed in L38-72/0, and 1809 were consistently suppressed both in L38-24/0 and L38-72/0 during *V. dahliae* inoculation (Figure 2C). Furthermore, the foldchange of these twelve groups was calculated. Most lgFoldChange of DEGs were between 1 to 10 and less number were more than 10 (Figure 2D).

### 3.4. Functional Analysis of Differentially Expressed Genes in L38 during V. dahliae Inoculation

To reveal potential functions of differentially expressed genes (DEGs), we identified the Gene Ontology (GO) terms of these genes. Therefore, the DEGs were divided into six groups for further analysis, including those specifically expressed in J1-24/0 (J1-Early), specifically expressed in J1-72/0 (J1-Late), expressed both in J1-24/0, J1-72/0 (J1-Sustained), expressed only in L38-24/0 (L38-Early), expressed only in L38-72/0 (L38-Late), and expressed both in L38-24/0 and L38-72/0 (L38-Sustained). Thus, 701, 899, 1422, 3524, 390, and 2078 DEGs matching to the above-mentioned groups were assigned GO terms. The regulation of macromolecule metabolic process and regulation of RNA metabolic process were highly enriched both in cultivar J1 and L38 (Figure 3A). Each group of L38 was higher than the corresponding group in J1, and the early and sustainably expressed DEGs were notably higher than the late expressed DEGs (Figure 3A). In addition, the DNA-binding transcription factor activity was also mainly enriched (Figure 3A). The enrichment of response to auxin, response to abiotic stimulus, response to oxidative stress, protein serine/threonine activity and peroxidase activity in J1 were higher than in L38 (Figure 3A). Interestingly, DEGs of J1 and L38 were equally enriched in anion transmembrane transporter activity (Figure 3A).

In addition, KEGG enriched pathways from each group were analyzed, and more than 100 pathways were assigned in each group, including metabolic pathways, biosynthesis of secondary metabolites and so on (Appendix A). Several pathways were related to resistance, such as plant-pathogen interaction, plant hormone signal transduction, MAPK signaling pathway, oxidative phosphorylation (Figure 3B and Appendix A). The number of genes in each group of L38 was higher than in J1, except for pyruvate metabolism (Figure 3B). Correspondingly, several studies have proved that genes related to protein serine/threonine kinase, peroxidase enzyme activity, plant-hormone signal transduction, response to biotic and abiotic stress, and lignin metabolism play a critical role in cotton disease resistance [34]. Above all, these results suggested that several DEGs of J1 and L38 cultivars were related to cotton resistance, with the DEGs of L38 enhancing resistant function more than J1 overall.

### 3.5. Resistance Related Pathways Analysis of DEGs

Among these GO terms and KEGG pathways, several DEGs were classified in resistance function. Therefore, the typical resistance pathway was further identified, including plant-pathogen interaction, plant hormone signal transduction, MAPK signaling pathway.

Plant–pathogen interaction was directly involved in plant resistance to pathogens. In the plant-pathogen interaction pathway, 369 genes can assign to 15 crucial points, including calcium-dependent protein kinase (*CDPK*), flagellin-sensing 2 (*FLS2*), *WRKY* transcription factor, *avrPphB-susceptible 1* (*PBS1*), heat shock protein 90 (*HSP90)*, and so on (Figure 4A). The associated information for these DEGs is shown in Appendix A. In PAMP-triggered immunity, DEGs in all groups of J1 and L38 assigned in *FLS2* and *Calmodulin and calmodulin-like* (*CaMCML*), but only DEGs of L38 assigned in MAPK/ERK kinase kinase (*MEKK1*) and MAP kinase (*MPK4*) (Figure 4A). In effector-triggered immunity, DEGs in all groups of J1 and L38 assigned in *RPS2*, and only DEGs of L38 assigned in RPM1 interacting protein 4 (*RIN4*) and *PBS1* (Figure 4A). Particularly, 245 DEGs were unique expressed in L38 and 57 DEGs were unique expressed in J1 (Appendix A). Moreover, expression pattern of five typical DEGs in plant-pathogen interaction were randomly selected to confirm by RT-qPCR, which confirmed the involvement of these genes in the response of L38 under inoculation with *V. dahliae* (Figure 4B).

Plant hormones involved in cell signaling impact defense responses and development have been proved may provide useful solutions incorporating aspects of basal defense [37]. In plant hormone signal transduction pathway, a total of 236 DEGs were assigned to hormone signaling of auxin (AUX), ET, JA, and SA (Appendix A). In detail, our analysis revealed that 92 DEGs are involved in regulation of auxin signal transduction, 53 DEGs are involved in ethylene signal transduction, 58 DEGs are involved in regulating JA signal transduction, while 33 DEGs were found to be involved in regulating SA signal transduction (Appendix A). In the auxin signal transduction pathway, DEGs only expressed in L38 were enriched in *TIR1*; in ethylene signal transduction, DEGs only expressed in L38 were assigned in *ETR*; in jasmonic acid signal transduction, DEGs uniquely expressed in L38 related to *JAZ*; in salicylic acid signal transduction, DEGs uniquely expressed in L38 regulated non-expression of pathogenesis-related gene 1 (*NPR1*) and pathogenesis-related gene 1 (*PR1*) (Figure 5A). To further confirm these findings, the expression level of five important genes were confirmed through qRT-PCR analysis, which showed that the hormone signaling is involved in the resistance response of L38 to *V. dahliae* (Figure 5B).

MAPK cascades are highly conserved signaling modules which play a pivotal role in signaling plant defense against pathogen attack by transducing extracellular stimuli into intracellular responses in the plant [38]. In the MAPK signal pathway, 255 DEGs were detected as regulating pathogen infection (flagellin) and pathogen attack (Appendix A). *Suppressor of mkk1 mkk2 2* (*SUMM2*), that is involved in reducing cell death defense response, was the only enriched DEG of J1 in the early stage which was different from other pathways (Figure 6A). Furthermore, the points of *MEKK1, MEKK1/2, MPK4*, MAP kinase substrate 1 (*MKS1*), and *PR1* were the only enriched DEGs of L38 (Figure 6A). In particular, only DEGs of L38 were assigned to *MEKK1* and *MPK4* when under H_2_O_2_ attack, which influences the accumulation of ROS during plant defense response, instead of DEGs of J1 enrichment (Figure 6A). Otherwise, the specific DEGs regulated by the MAPK signal pathway were selected and the expression pattern was confirmed by RT-qPCR analysis (Figure 6B).

### 3.6. Discovering and Characteristic of Resistance Gene Analogues (RGAs) in J1 and L38

Resistance (*R*) genes have been shown to play an essential role in recognizing effectors from pathogens and in triggering downstream signaling during plant disease resistance [17,39]. The resistance gene analogues (RGAs) carry the specific domain the same as *R* genes, including nucleotide-binding sites (NBSs), leucine-rich repeat (LRR), Toll/Interleukin-1 receptor (TIR), coiled-coil (CC) and transmembrane (TM) [17]. Therefore, we focused on the RGAs carrying the NBS, LRR, TIR, CC, and TM domain in J1 and L38 transcriptome following *V. dahliae* inoculation. A total of 486 RGAs were identified from six groups, including J1-early, J1-late, J1-sustained, L38-early, L38-late, and L38-sustained. 358 RGAs were identified in L38, significantly higher than in J1 which displayed only 128 such RGAs (Figure 7A). Moreover, there were 195 RGAs in L38-late group which was the highest level among all groups while 130 RGAs were consistently expressed in L38 (Figure 7A). A total of 14 RGAs were identified as common among six groups (Figure 7B). Interestingly, the expression pattern of these 14 RGAs was opposite between J1 and L38, such as the expression of *LOC107902722* (NBS-LRR), which was down regulated in J1 while up regulated in L38 (Figure 7B). For these 14 RGAs, there were ten genes down-regulated and four genes up-regulated in the susceptible J1 cultivar, in a contrasting manner to that observed (four genes up-regulated and ten genes down-regulated) in the L38 resistant cultivar (Figure 7B). In addition, the expression level of RGAs were analyzed by FPKM values. The heat map showed that RGAs were divided into three groups (Figure 7C). RGAs of group 1 expressed highly in L38, and RGAs of group 2 highly expressed in J1 (Figure 7C). However, majority of RGAs highly expressed both in J1 and L38 belong to group 3 (Figure 7C). Furthermore, the foldchange of these six groups was calculated. Most lgFoldChange of DEGs were between 1 to 10 and only a few candidates were more than 10 (Figure 7D). Thus, the widely and strongly observed response of RGAs should play a critical role in the resistant of L38 compared to the susceptible cultivar of J1 under inoculation with *V. dahliae*.

## 4. Discussion

Plants have evolved an effective and complicated immune system to resist pathogen attack. Cotton is one of the most economically important crops. VW is a serious disease of cotton, leading to significant decrease in the yield of cotton. In recent years, several resistant genes have been identified in cotton which are involved in VW resistance. The molecular function mechanism of cotton resistance to *V. dahliae* is becoming clearer, thus assisting in the breeding of resistant cotton cultivars. In fact, the gene regulation network underlying the molecular mechanisms leading to resistance seen in several cotton cultivars with excellent VW tolerance is relatively unexplored. For instance, the cultivar Liaomian bred from the Research of Economic Crops in Liaoning Province showed higher resistance during production, but the exact mechanism underlying its resistant function remained unclear. In this study, we first confirmed the resistance phenotype of L38 compared with VW susceptible cotton J1 (Figure 1). Previous work has shown that differential expression and regulation of resistance-related genes played an important role in plant response processes to various pathogens [21,40,41]. Therefore, we conducted RNA-seq analysis to identify the DEGs in L38 response to *V. dahliae* infection. A number of DEGs were observed by comparing the transcriptome of these two cultivars at various timepoints (J1-24/0, J1-72/0, L38-24/0, and L38-72/0). These results show that the expression pattern of genes in L38 and J1 respond differentially to *V. dahliae* infection (Figure 2B,C). There were common as well as unique DEGs among the four different comparison groups. The unique DEGs in J1 and L38 were chosen for further analysis and divided into six groups based on the timeline of their expression during *V. dahliae* infection, including J1-Early, J1-Late, J1-Sustained, L38-Early, L38-Late, and L38-Sustained. The number of unique DEGs in L38 was higher than in J1, suggesting that more genes related to resistance in L38 cultivar are induced thus providing timely defense response against *V. dahliae* infection. Furthermore, the potential function of these DEGs have been analyzed by GO enrichment and KEGG pathway. It is notable that many DEGs were related to resistance, including several DEGs significantly enriched in response to auxin, oxidative stress and abiotic stimulus, protein serine/threonine, and peroxidase activity (Figure 3A). Additionally, many DEGs were also enriched in resistance related pathways, such as plant-pathogen interaction, plant hormone signal transduction, MAPK signaling pathway, and oxidative phosphorylation (Figure 3B, Figure 4, Figure 5 and Figure 6 and Appendix A). Expression of several specific DEGs were up-regulated or down-regulated and were confirmed further by qRT-PCR analysis, indicating that these genes might play a greater role in resistance response to *V. dahliae* infection. Furthermore, RGAs which potentially encapsulate a large cohort of putative *R* genes that display conserved domains and common structural features may provide candidate genes for breeding enhanced VW tolerance in cotton [41]. In our current study, DEGs involved in LRR, NBS-LRR, TIR, CC, and TM domain were considered as RGAs. A total of 486 such RGAs were identified from J1 and L38 cotton cultivars that respond differentially to *V. dahliae* infection. There were 358 such RGAs in the resistant L38 cultivar, significantly higher in number than in the susceptible J1 cultivar with only 128 RGAs (Figure 7A). For J1, 54 RGAs expressed at 24 hpi and 49 RGAs at 72 hpi, which was nearly the same for both; for L38, 33 RGAs expressed at 24 hpi and 195 RGAs at 72 hpi. The number of RGAs expressed by L38 at 72 hpi comprises most of these RGAs. These results suggest the efficiency response of *R* genes should play a critical role in disease resistance, thus contributing to the Verticillium wilt resistance in L38.

*R* genes can act as guardians to monitor the modification of host proteins after associating with the pathogenic effectors, resulting in the initiation of resistance, such as flagellin-sensitive 2 (*FLS2*), *RPM1 interacting 4* (*RIN4*), *avrPphB susceptible 1* (*PBS1*), and *Resistance to Pseudomonas syringae 5* (*RPS5*) [42]. From our data, the DEGs from J1 and L38 after early and late inoculation were all enriched in *FLS2* in plant-pathogen interaction pathway, but DEGs induced in L38 were assigned to *RIN4* and *PBS1* related pathways (Figure 4A). In addition, DEGs of L38 were uniquely assigned to *MEKK1* and *MPK4* pathways, that contribute to pathogen induced accumulation of ROS during plant defense response (Figure 4A and Figure 6A). These results suggested that though expression of several genes in cotton is significantly induced against *V. dahliae* in general, some of these genes are uniquely expressed in the resistant L38 cultivar during *V. dahliae* infection. Our finding indicates that some of these specifically induced resistance-related genes could be potential candidates in fortifying the observed resistance in the L38 cultivar to *V. dahliae* infection. Additionally, several DEGs in our analysis that were assigned to plant hormone regulation are the key point of auxin, ETH, JA, and SA signal transduction pathways which are known to play a role in VW resistance in host plant species (Figure 5A). These results add further validation to the growing body of evidence for the role of such hormone signaling pathways in Verticillium resistance [37], thus indicating related mechanisms that underscore the increased resistance of the L38 cultivar.

However, RGAs are important regulators in the plant resistance response to various pathogens. In general, RGAs can be classified into nucleotide binding site leucine rich repeat (NBS-LRR) and transmembrane leucine rich repeat (TM-LRR). More RGAs were significantly expressed in L38 than in J1 following *V. dahliae* inoculation. Moreover, most of RGAs in the resistant L38 cultivar were different from those in the susceptible J1 cultivar. Interestingly, 14 such RGAs that were down-regulated in J1 were up-regulated in the resistant L38 cultivar (Figure 7A,B). This result presents yet another sector of defense response in L38 that is markedly different following *V. dahliae* infection. Taken together, our results have identified various sectors of the cotton transcriptome connecting network of genes coordinately expressed in the resistant cotton cultivar L38 that might provide a basis for the observed resistance against *V. dahliae* in contrast to the susceptible J1 cultivar.

In short, DEGs of cotton with a putative role in VW resistance were discovered by RNA-seq analysis of two cotton cultivars differing in their resistance to *V. dahliae*. Many DEGs were found through subsequent GO and KEGG analysis that related to function of VW resistance in cotton. The expression level of candidate DEGs were reconfirmed to ensure the regulation of these genes related to defense response upon *V. dahliae* inoculation. Moreover, the cotton RGAs were also analyzed during *V. dahliae* inoculation revealing that RGAs in the resistant L38 cultivar were typically different from those in the susceptible J1 cultivar, implying that the regulation network of resistance-related genes in L38 was more novel and diverse than in the susceptible cotton cultivar. This study lays a solid foundation for further in-depth analysis of myriad of candidate resistance genes in the VW resistant L38 cotton cultivar. At present we are further exploring the functional mechanism of typical resistance-related genes revealed in our study presented here to elucidate the molecular basis of VW resistance observed in the resistant L38 cotton cultivar.

## 5. Conclusions

In conclusion, it is our understanding that the inter-connected regulatory network of these gene clusters in the resistant cotton cultivar L38 upon pathogen infection provides a novel and comprehensive insight into the mechanism of VW resistance in pest resistant cotton cultivars. The findings of this study can be harnessed to design increased tolerance to VW in cotton, which is currently under significant threat from this devastating disease worldwide.

## Figures and Tables

**Figure 1 cells-10-02961-f001:**
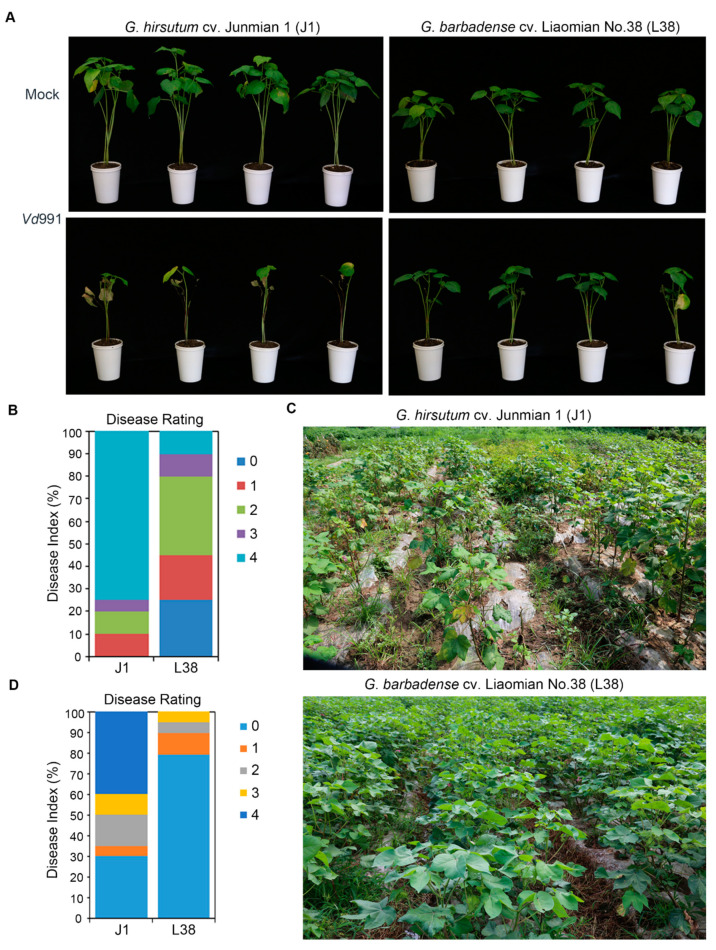
Phenotype of *G**ossypium hirsutum* cv. Junmian No.1 (J1) and cv. Liaomian No. 38 (L38) cotton defense against *Verticillium dahliae*. (**A**) Disease symptoms of J1 and L38 cotton following inoculation with *V. dahliae* strain Vd991 and treatment with water as mock/control. The severity of cotton wilting as shown in the corresponding pictures recorded at 14 days post inoculation (dpi). (**B**) Disease index of J1 and L38 cotton were determined at 14 dpi after Vd991 inoculation. Disease grade 0, 1, 2, 3, and 4 showed the disease range from asymptomatic to lethal. (**C**) Disease symptoms of J1 and L38 cotton planted in disease nursery of Liaoyang city at July 2021. (**D**) Disease index of J1 and L38 cotton planted at the Liaoyang disease nursery was determined during July 2021. Disease grades 0, 1, 2, 3, and 4 encapsulate the disease from asymptomatic to lethal.

**Figure 2 cells-10-02961-f002:**
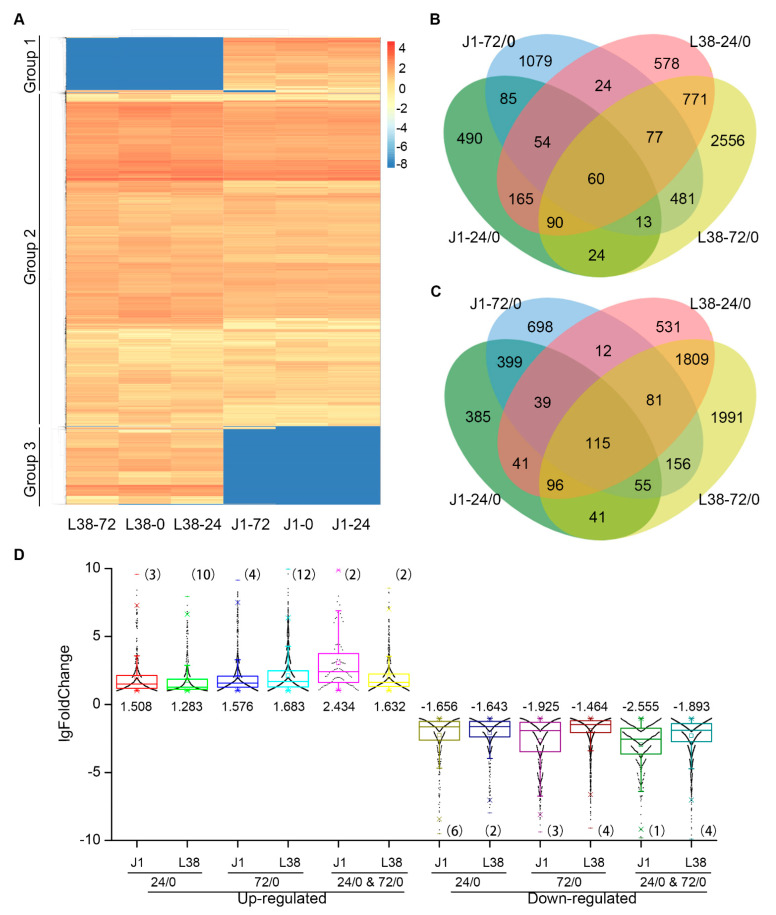
Analysis of differentially expressed genes (DEGs) in J1 and L38 cotton during *V. dahliae* inoculation. (**A**) Heatmap showed the time course expression profiles of genes annotated in J1 and L38 at 0 h, 24 h, and 72 h, respectively, which was performed by FPKM values of all genes. (**B**) The Venn diagram of the up-regulated DEGs indicated unique and common DEGs for four different comparisons, including J1-24 h/ 0 h (J1-24/0), J1-72/0 h (J1-72/0), L38-24 h/ 0 h (L38-24/0), and L38-72 h/ 0 h (L38-72/0). (**C**) The Venn diagram of the down-regulated differentially expressed genes (DEGs) indicated unique and common DEGs for four different comparisons, including J1-24 h/ 0 h (J1-24/0), J1-72/0 h (J1-72/0), L38-24 h/ 0 h (L38-24/0), and L38-72 h/ 0 h (L38-72/0). (**D**) The fold change of up- and down-regulated DEGs in J1 and L38 at 24/0, 72/0, and both 24/0 and 72/0 (24/0 & 72/0). The numbers indicate the median of all lgfoldchange of DEGs in each group, and the number in bracket indicates the number of DEGs for which lgfoldchange was higher than 10 or lower than −10.

**Figure 3 cells-10-02961-f003:**
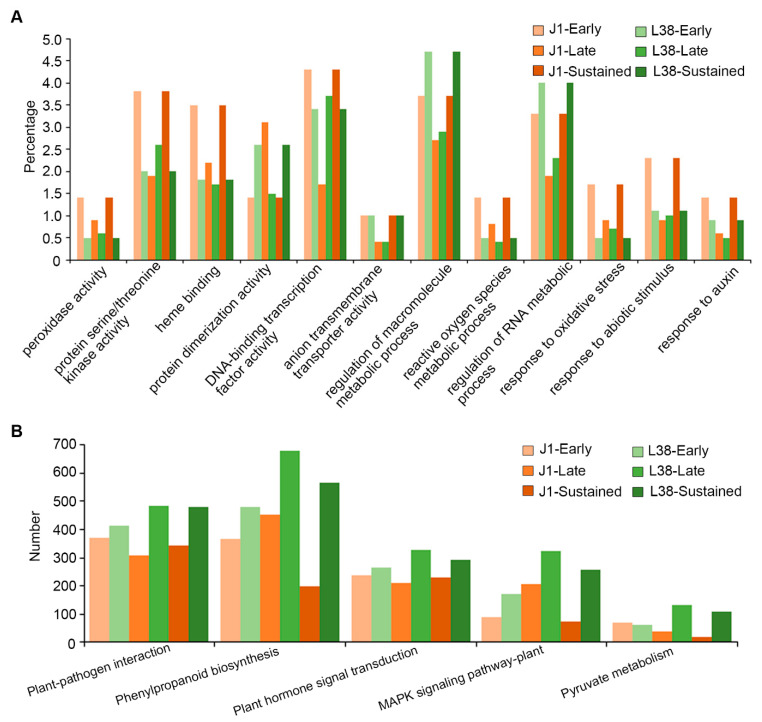
Gene ontology (GO) enrichment and KEGG pathway analysis of DEGs in J1 and L38 during *V**erticillium dahliae* inoculation. (**A**) GO analysis for these DEGs classified in six groups, with *p*-value < 0.05. (**B**) Pathways related to resistance were identified by KEGG analysis with *p*-value < 0.05.

**Figure 4 cells-10-02961-f004:**
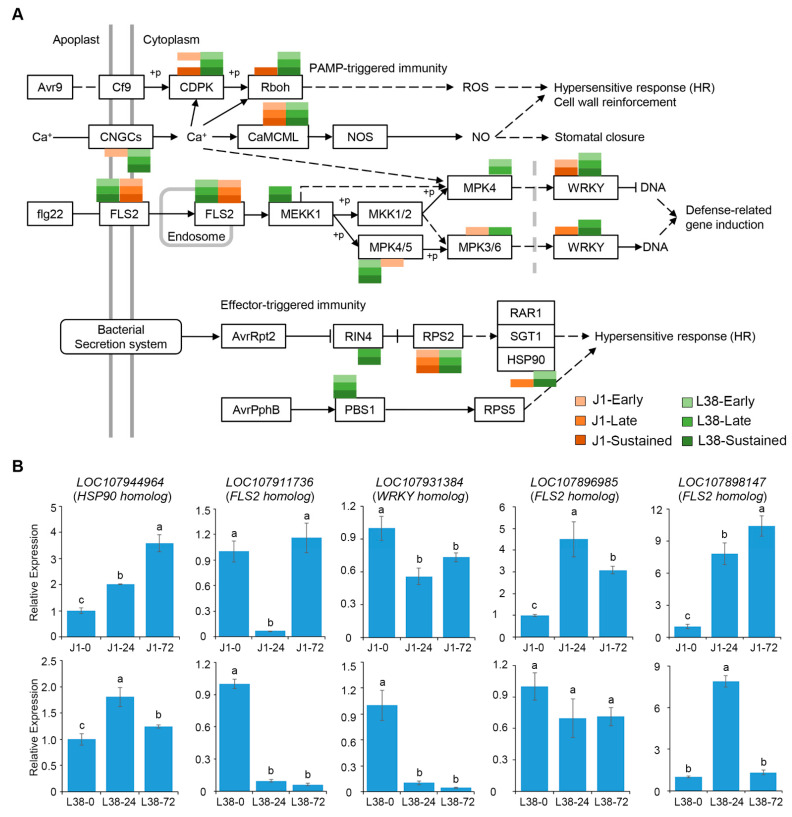
DEGs were assigned in plant-pathogen interaction. (**A**) Demonstration of enrichment of DEGs in plant-pathogen interaction with *p*-value < 0.05. (**B**) RT-qPCR analysis of expression pattern of specific DEGs of J1 and L38 assigned to plant-pathogen interaction cluster. Relative expression levels were normalized by cotton *18S* gene expression level. The relative gene expression level in J1-0 and L38-0 were assigned a value of 1.0. Error bars indicate ±SD of three biological replicates, with each measured in triplicate. Samples marked with different letters show a significant difference at *p* < 0.05 by Duncan’s multiple range test.

**Figure 5 cells-10-02961-f005:**
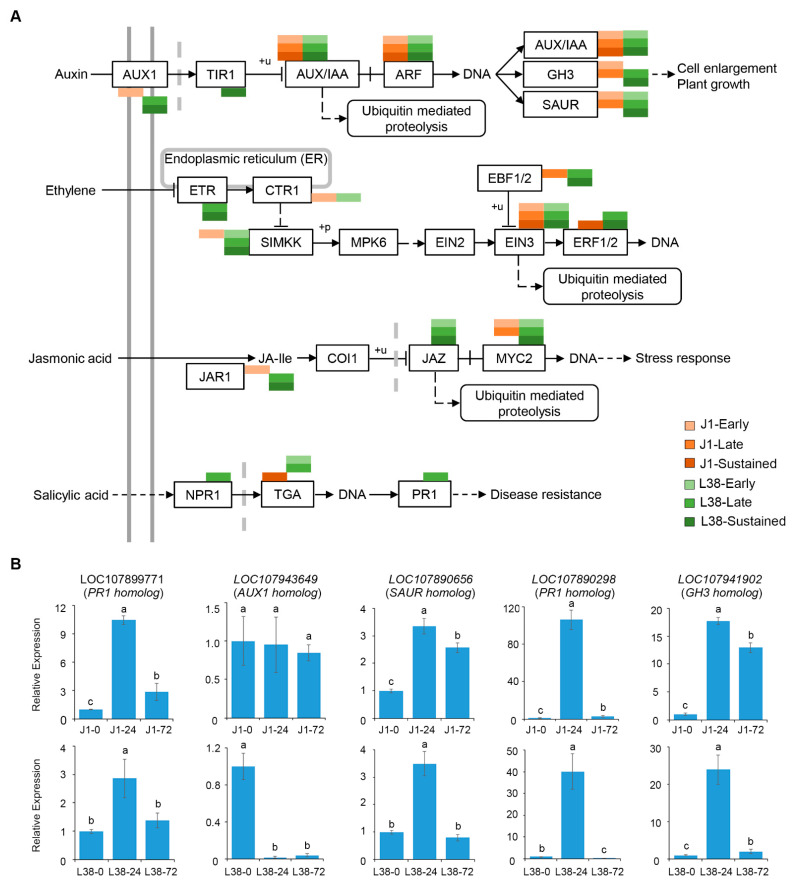
DEGs were enriched in plant hormone signal transduction. (**A**) Demonstration of enrichment of DEGs assigned in plant hormone signal transduction with *p*-value < 0.05. (**B**) Expression pattern analysis of specific DEGs of J1 and L38 in plant hormone signal transduction by qRT-PCR. Relative expression levels were normalized by cotton 18S expression level. The relative gene expression level in J1-0 and L38-0 were assigned a value of 1. Error bars indicate ±SD of three biological replicates, with each measured in triplicate. Samples marked with different letters show a significant difference at *p* < 0.05 by Duncan’s multiple range test.

**Figure 6 cells-10-02961-f006:**
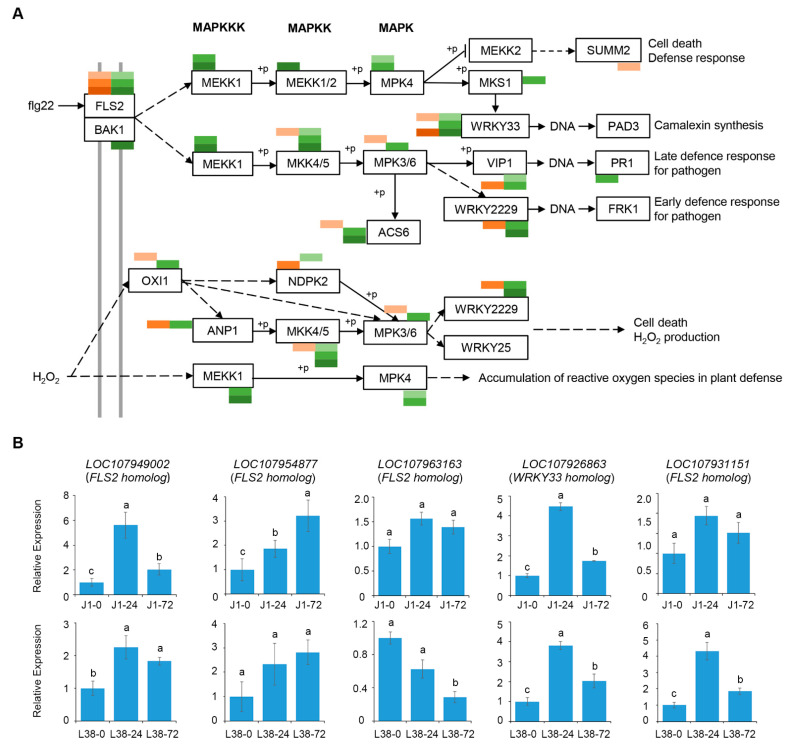
DEGs were enriched in MAPK signaling pathway. (**A**) Demonstration of enrichment of DEGs in MAPK signaling pathway with *p*-value < 0.05. (**B**) Expression pattern analysis of specific DEGs of J1 and L38 assigned in MAPK signaling pathway by RT-qPCR. Relative expression levels were normalized by cotton 18S expression level. The relative gene expression level in J1-0 and L38-0 were assigned a value of 1. Error bars indicate ±SD of three biological replicates, with each measured in triplicate. Samples marked with different letters show a significant difference at *p* < 0.05 by Duncan’s multiple range test.

**Figure 7 cells-10-02961-f007:**
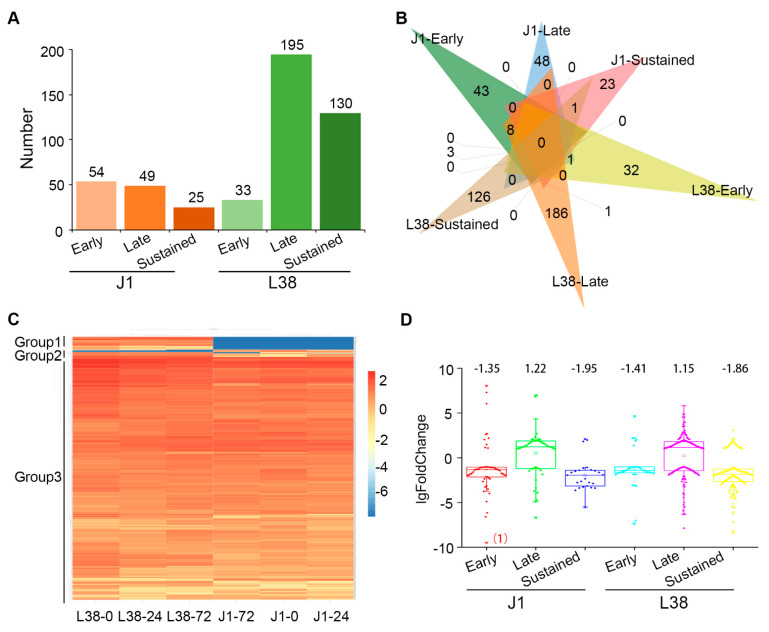
Characteristic of resistance gene analogues (RGAs) in J1 and L38 during *Verticillium dahliae* inoculation. (**A**) Number of RGAs in J1-early, J1-late, J1-sustained, L38-early, L38-late, and L38-sustained, respectively. (**B**) Venn diagram showed RGAs that were commonly and uniquely differentially expressed in J1-early, J1-late, J1-sustained, L38-early, L38-late, and L38-sustained, respectively. (**C**) Heatmap showed the time course expression profiles of RGAs annotated in J1 and L38 at 0 h, 24 h, and 72 h, respectively, which was performed by FPKM values of all genes. (**D**) The fold change of RGAs in J1-early, J1-late, J1-sustained, L38-early, L38-late, and L38-sustained. The number indicates the median of all log fold change of DEGs in each group, and the number in brackets indicates the number of RGAs for which log fold change was than −10.

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
