# Peer review of "Transcriptome Analysis of a Cotton Cultivar Provides Insights into the Differentially Expressed Genes Underlying Heightened Resistance to the Devastating Verticillium Wilt"

_cells, 2021, doi:10.3390/cells10112961_

Round 1

Reviewer 1 Report

This manuscript provides insights into the resistance mechanisms of the resistant to VW L38 cultivar of cotton in comparison to the susceptible J1 cultivar, using transcriptomic analyses.

The authors initially showed that, as expected, the L38 cultivar is resistant to VW in comparison to the J1 cultivar. Then they proceed to RNA-seq analysis at 24 and 72h post inoculation and analyzed the differentially expressed genes.

The manuscript is well written, with only minor spelling and expression issues, and the methods are up to date with state of the field. The conclucions drawn by the authors are supported by the data provided.

Some minor issues in the manuscript that need to be more clearly stated or presented

-this study used RNA from inoculated roots only, you cite [12] but you could provide a short explanation of why this is enough (meaning why not RNA from other tissues is not necessary)

-Figure 3A is not easy to read, maybe seperate the categories a little better (they need more space between the bars)

-at paragraph 3.4 you state "The most
enrichment category was metabolic process, including regulation of macromolecule metabolic process and regulation of RNA metabolic process..." that is not entirely true since reactive oxygen species category is not very enriched - you should rephrase this part

-later at paragraph 3.4 you should explain why RNA metabolism and macromolecule metabolism are considered resistance related functions, i understant that resistance related functions could be included in these categories but these are very broad categories of biological functions and a short explaination would benefit the manuscript

-at paragraph 3.5 you state that 369 genes regulate 15 crucial points including CDPK, FLS2... are the crucial genes CDPK, FLS2 included in this analysis?

-also at paragraph 3.5 the q-RT-PCR-analyzed genes were randomly selected from each category or there was a logic behind the selections?

the codes for the genes do not help to understand their meaning or function and

finally wouldn't be helpful to compare the expression levels of the L38 and J1 cultivars? in the figures 4B, 5B and 6B the expression is adjusted to 1 for the 0h time point for each cultivar but this does not help to compare the expression levels for the two cultivars

i believe the manuscript would benefit if the paragraphs 3.4 and 3.5 rephrased, and better explain the logic behind the experiments and analyses presented

-At Discussion you state " These results probably proved that
R genes function at the second field of immune.
" the word prove is a bold claim, better use the words strongly suggest or something similar

Author Response

Reviewer 1

This manuscript provides insights into the resistance mechanisms of the resistant to VW L38 cultivar of cotton in comparison to the susceptible J1 cultivar, using transcriptomic analyses.

The authors initially showed that, as expected, the L38 cultivar is resistant to VW in comparison to the J1 cultivar. Then they proceed to RNA-seq analysis at 24 and 72h post inoculation and analyzed the differentially expressed genes.

The manuscript is well written, with only minor spelling and expression issues, and the methods are up to date with state of the field. The conclusions drawn by the authors are supported by the data provided.

Some minor issues in the manuscript that need to be more clearly stated or presented

Question 1

-this study used RNA from inoculated roots only, you cite [12] but you could provide a short explanation of why this is enough (meaning why not RNA from other tissues is not necessary)

Answer 1

Thank you for your valuable comments that also referred by Reviewer 2. Verticillium dahliae was a soil-borne pathogen which attack host plants through root. The fungus takes around 2 ~ 4 d to reach the xylem vessels and proliferates in the vascular. Disease symptoms may comprise wilting, chlorosis, stunting, necrosis, vein clearing and brown vascular discoloration may be observed in stem tissue cross-sections (Fradin and Thomma, Molecular Plant Pathology, 2006). Therefore, we employed the roots inoculation samples for transcriptome analysis for the specific biological characteristics of V. dahliae. Actually, almost of the transcriptome analysis of cotton-Verticillium were employed the root inoculation method. We added the explanation in revised manuscript.

Question 2

-Figure 3A is not easy to read, maybe separate the categories a little better (they need more space between the bars)

Answer 2

Thank you for your comment. We corrected it.

Question 3

-at paragraph 3.4 you state "The most enrichment category was metabolic process, including regulation of macromolecule metabolic process and regulation of RNA metabolic process..." that is not entirely true since reactive oxygen species category is not very enriched - you should rephrase this part

-later at paragraph 3.4 you should explain why RNA metabolism and macromolecule metabolism are considered resistance related functions, I understand that resistance related functions could be included in these categories but these are very broad categories of biological functions and a short explanation would benefit the manuscript

Answer 3

We have modified this part according to your suggestion, and added explanation and references in revised manuscript.

Question 4

-at paragraph 3.5 you state that 369 genes regulate 15 crucial points including CDPK, FLS2... are the crucial genes CDPK, FLS2 included in this analysis?

Answer 4

We are sorry for unclear clarify. The 369 genes assigned in 15 crucial points such as CDPK, FLS2 based on their accession number of KEGG database. There were 111 genes assigned in FLS2, and 27 genes in CDPK.

Question 5

-also at paragraph 3.5 the q-RT-PCR-analyzed genes were randomly selected from each category or there was a logic behind the selections?

Answer 5

Genes in qRT-PCR were randomly selected from each category to confirm the reliability of the transcriptome data.

Question 6

the codes for the genes do not help to understand their meaning or function and finally wouldn't be helpful to compare the expression levels of the L38 and J1 cultivars? in the figures 4B, 5B and 6B the expression is adjusted to 1 for the 0 h time point for each cultivar but this does not help to compare the expression levels for the two cultivars. I believe the manuscript would benefit if the paragraphs 3.4 and 3.5 rephrased, and better explain the logic behind the experiments and analyses presented.

Answer 6

Thank you for this helpful suggestion.

Firstly, we added the gene names to make the meaning is easy to understand.

Secondly, we mainly focused on the reliable of RNA-seq and the expression pattern tendency of genes in J1 and L38 in separate, to showed that the different response pattern between susceptible and resistance cultivar. Therefore, the expression is adjusted to 1.0 for the 0 h time point for each cultivar.

Finally, we have rephrased paragraph 3.4 and 3.5 and added some information according to your suggestion in the revised manuscript.

Question 7

-At Discussion you state " These results probably proved that genes function at the second field of immune. " the word prove is a bold claim, better use the words strongly suggest or something similar

Answer 7

Thank you for your valuable comment. We corrected it.

Reviewer 2 Report

The present study investigates transcriptome changes in response to pathogen infection in pathogen-resistant cotton cultivar, L38, and susceptible cultivar, J1. The authors confirm the difference in pathogen resistance between the two cultivars and performed RNA-seq at two time-points of pathogen infection in addition to the control time-point (0 h). Authors identified differential gene expression analysis in each genotype at each time point, revealing thousands of DEGs. GO terms analysis revealed significant enrichment of cellular signaling pathways including pathogen, plant hormone, and MAPK signaling pathways, Authors further reported expression patterns of the resistance gene analogs (RGAs) in the RNA-seq datasets. Here are my comments. Since there are no line numbers, it was difficult for me to point the exact location in the texts regarding each comment. 

Major issues
1) The authors stated the main question in the introduction: “the molecular mechanisms underlying the excellent resistance of Liao cotton to VW has not been clearly defined and hence is poorly understood. Therefore, in the current work, we conducted RNA-seq…” I do not think this study can address the main question. This study profiled transcriptome, but there were functional characterization and network analyses. No molecular mechanism was revealed in this study. 
2) Concerning comment #1, the title needs to be modified. For example, authors should not use “gene regulatory network” because there is no such thing in this study. 
3) Figure 1 shows phenotypes in shoots, but RNA-seq was done in roots. Authors need to justify why the RNA-seq was done in roots. 
4) Figure 2A and Figure 7C: what is the unit of the heatmap scale? How was the grouping achieved? Provide the hierarchical clustering tree. 

Minor issues:
1) Method 2.2 (RNA extraction, library prep) needs to be improved. What library prep kit was used? Is it strand-specific? 
2) Authors should provide SRA or GEO accession numbers with a token for reviewers. 
3) Method 2.3 (identification of DEGs): what program/version was used to calculate FPKM?
4) Method 2.4 (qRT-PCR): what instrument was used for qRT-PCR?

Author Response

Reviewer 2

The present study investigates transcriptome changes in response to pathogen infection in pathogen-resistant cotton cultivar, L38, and susceptible cultivar, J1. The authors confirm the difference in pathogen resistance between the two cultivars and performed RNA-seq at two time-points of pathogen infection in addition to the control time-point (0 h). Authors identified differential gene expression analysis in each genotype at each time point, revealing thousands of DEGs. GO terms analysis revealed significant enrichment of cellular signaling pathways including pathogen, plant hormone, and MAPK signaling pathways, Authors further reported expression patterns of the resistance gene analogs (RGAs) in the RNA-seq datasets. Here are my comments. Since there are no line numbers, it was difficult for me to point the exact location in the texts regarding each comment. 

Major issues

Question1

  • The authors stated the main question in the introduction: “the molecular mechanisms underlying the excellent resistance of Liao cotton to VW has not been clearly defined and hence is poorly understood. Therefore, in the current work, we conducted RNA-seq…” I do not think this study can address the main question. This study profiled transcriptome, but there were functional characterization and network analyses. No molecular mechanism was revealed in this study. 

Answer 1

Thank you for your grateful comment. We have changed this part in the revised manuscript.

Question 2

2) Concerning comment #1, the title needs to be modified. For example, authors should not use “gene regulatory network” because there is no such thing in this study. 

Answer 2

Thank you for your grateful comment. We have changed this part in the revised manuscript.

Question 3

3) Figure 1 shows phenotypes in shoots, but RNA-seq was done in roots. Authors need to justify why the RNA-seq was done in roots. 

Answer 3

Thank you for your valuable comments that also referred by Reviewer 1. Verticillium dahliae was a soil-borne pathogen which attack host plants through root. The fungus takes around 2 ~ 4 d to reach the xylem vessels of the root, and proliferates in the vascular. Disease symptoms may comprise wilting, chlorosis, stunting, necrosis, vein clearing and brown vascular discoloration may be observed in stem tissue cross-sections (Fradin and Thomma, Molecular Plant Pathology, 2006). Therefore, we employed the roots inoculation samples for transcriptome analysis for the specific biological characteristics of V. dahliae. Actually, almost of the transcriptome analysis of cotton-Verticillium were employed the root inoculation method. We added the explanation in revised manuscript.

Question 4

4) Figure 2A and Figure 7C: what is the unit of the heatmap scale? How was the grouping achieved? Provide the hierarchical clustering tree. 

Answer 4

The unit of the heatmap scale is log10 (FPKM value). We added the hierarchical clustering tree in heatmap. The groups were obtained based on hierarchical clustering tree. We have changed them in revised manuscript.

Minor issues:

Question 5

1) Method 2.2 (RNA extraction, library prep) needs to be improved. What library prep kit was used? Is it strand-specific?

Answer 5

Thank you for your suggestion. The library prep kit was Collibri™ Stranded RNA Library Prep Kit for Illumina™ Systems (Thermo, USA), and the sequencing was strand-specific. We have modified this.

Question 6

2) Authors should provide SRA or GEO accession numbers with a token for reviewers. 

Answer 6

The data of clean reads has been uploading. We would like to provide SRA accession number as soon as possible, to cover the requirement from journal.

Question 7

3) Method 2.3 (identification of DEGs): what program/version was used to calculate FPKM?

Answer 7

Bowtie2 was used to map clean reads into reference genome, and then RSEM was used to calculate the value of FPKM. We added this information in revised draft.

Question 8

4) Method 2.4 (qRT-PCR): what instrument was used for qRT-PCR?

Answer 8

Correct it.

Reviewer 3 Report

Dear Authors,

                I have a great honor and opportunity to review scientific manuscript entitled: “Transcriptome Analysis of a Cotton Cultivar Provides Insights into the Gene Regulatory Networks Underlying Heightened Resistance to the Devastating Verticillium wilt” which is considered for publication in Cells journal. Article is interesting and present interesting new insight into problem of resistance to the Verticillium wilt. I analyzed whole manuscript and it is interesting however in some points English need improvement by native speaker. Nevertheless, the manuscript is good in research level and need some improvements in a form of research presentation and criteria explanation. I presented my suggestion in a form of list present below:

  1. Introduction section:

I would like kindly ask authors to formulated more precisely aim of the study. With use sentence “Aim of the study was”.  

  1. Results

This section is little problematic because of presents of Figures overloaded with data

Figure 1 must be split to separate figures now the A and C parts are too small to clearly observed the results. All photos of Disease symptoms must big enough to be clearly observed. Moreover, at Figure 1A all type of diseases symptoms must be named and physically marked on photos  with use of arrows asterisk or other markings. This part must have all needed markings it will be also better if on photos with symptoms more directly show the leaf itself The statement ‘disease symptoms” without precision terminology and markings is illogical. Moreover, Figure 1B I am sorry but because the size and form of photos the differences (which they are not named at all) between susceptible and resistant plants is extremely difficult to observe. I suggest to change photos or enlarge them. I would like also to kindly ask to explain criteria used to assessment of results presented on Figure 1B and Figure 1D. Because numerical data should be analyzed statistically on the first step of analyzes.

Figure2 must be split to separate Figures and enlarge as big as it is possible. Now they are too low quality because their small size to be readable.

Figure 5, 6,7 and 8 must be split to separate Figures. Now they are too low quality because their small size to be readable.

Minor

I also suggest to change font size of references list because now it is much larger than main text of article.

Author Response

Reviewer 3

 I have a great honor and opportunity to review scientific manuscript entitled: “Transcriptome Analysis of a Cotton Cultivar Provides Insights into the Gene Regulatory Networks Underlying Heightened Resistance to the Devastating Verticillium wilt” which is considered for publication in Cells journal. Article is interesting and present interesting new insight into problem of resistance to the Verticillium wilt. I analyzed whole manuscript and it is interesting however in some points English need improvement by native speaker. Nevertheless, the manuscript is good in research level and need some improvements in a form of research presentation and criteria explanation. I presented my suggestion in a form of list present below:

Question 1

  1. Introduction section:

I would like kindly ask authors to formulated more precisely aim of the study. With use sentence “Aim of the study was”.

Answer 1

We modified it in resubmitted manuscript according to your suggestion. Thank you!

Question 2

  1. Results

This section is little problematic because of presents of Figures overloaded with data

Question 3 

Figure2 must be split to separate Figures and enlarge as big as it is possible. Now they are too low quality because their small size to be readable.

Question 4

Figure 5, 6,7 and 8 must be split to separate Figures. Now they are too low quality because their small size to be readable.

 Answer 2

Thank you for your suggestion. We would like to answer these three questions together:

1) We added the necessary elements, such as hierarchical clustering tree to complete the information of figures;

2) We reformate the figures to improve the image resolution according your valuable suggestions;

3) We magnified words in figures to improve the image resolution as we can;

4) We would like to show one result section with one corresponding figure, to make the results in logical and easy for reader to follow, so we apply to maintain the structure of figure as present version;

5) In addition, we would like to supply the high-quality figures with TIF format to satisfy the journal requirement for publishing.

 Question 5

Figure 1 must be split to separate figures now the A and C parts are too small to clearly observed the results. All photos of Disease symptoms must big enough to be clearly observed. Moreover, at Figure 1A all type of diseases symptoms must be named and physically marked on photos with use of arrows asterisk or other markings. This part must have all needed markings it will be also better if on photos with symptoms more directly show the leaf itself. The statement ‘disease symptoms” without precision terminology and markings is illogical. Moreover, Figure 1B I am sorry but because the size and form of photos the differences (which they are not named at all) between susceptible and resistant plants is extremely difficult to observe. I suggest to change photos or enlarge them. I would like also to kindly ask to explain criteria used to assessment of results presented on Figure 1B and Figure 1D. Because numerical data should be analyzed statistically on the first step of analyzes.

Answer 3

Thank you for your comment. We modified Figure 1 according to your suggestion.

Question 6

Minor

I also suggest to change font size of references list because now it is much larger than main text of article.

Answer 4

We are sorry for this mistake. We corrected font size of references list in new manuscript.

Round 2

Reviewer 2 Report

In the revision of their manuscript, Zhu et al address the issues which were raised before, carefully responding to my comments. However, my request for the accession number of the RNA-seq data is still not addressed. Uploading sequencing data to GEO generally takes less than a week. As a peer researcher, I would like to ask the editor to ensure those data is available to the communities upon accepting this study for publication. 

Reviewer 3 Report

Dear Authors,

Thank you for all kind asnwers. Manyscript is much improved so I recomend publication

Best regards,